# The Impact of Temperature Conditions on the Manufacturing Process and Mechanical Behavior of Beverage Can Ends during Operation

**DOI:** 10.3390/ma16186137

**Published:** 2023-09-09

**Authors:** Paweł Kokoszka, Andrij Milenin

**Affiliations:** 1CANPACK S.A., Business Support Service, Starowiejska 28 Street, 32-800 Brzesko, Poland; 2Faculty of Metals Engineering and Industrial Computer Science, AGH University of Krakow, Mickiewicza 30 Av., 30-059 Kraków, Poland; milenin@agh.edu.pl

**Keywords:** AA5182-H48 alloy, beverage end, stamping, aging, buckling, manufacturing process

## Abstract

This article investigates the impact of temperature regimes, corresponding to various climatic zones, on the manufacturing process and operation of beverage can ends made of aluminum alloy AA5182. The production process of aluminum beverage can ends involves multiple steps, including melting, rolling, and stamping, where different temperatures can influence both the production process and the properties of the final product. Furthermore, the mechanical behavior of the final product is affected by the aging process of alloy AA5182, which progresses at varying rates depending on the temperature conditions during storage. The objective of this study is to simulate the production process under various climatic conditions using the finite element method (FEM) and experimentally investigate the dependence of the strength of alloy AA5182 can ends on storage time and temperature. The findings of the study reveal that, under temperature conditions corresponding to warmer climates, the punching force can be reduced by approximately 15% compared to production in colder climates. Additionally, the strength of the finished can exhibits a decrease of about 10% during a month of storage in a warm climate, while no significant decrease was observed in colder climates. These results hold practical significance for the beverage production industry, as the manufacturing and operation of beverage cans are localized in diverse climatic zones.

## 1. Introduction

Beverage can ends as a part of cans’ closure are one of the most popular forms of packaging for carbonated drinks [1]. These ends are typically made of can end stock (CES) AA5182 alloys [2,3], which are lightweight, corrosion-resistant, and can be easily formed into the desired shape. Their primary purpose is to provide a secure closure and easy opening mechanism [4]. The production of aluminum ends involves several steps. Once the shell ends are shaped, they are converted on the press, where the tab is formed and attached to the shell, which undergoes additional forming operations. The shape and design of beverage ends have evolved over the years. Initially, ends had a flat top. Later, the pull-tab design was introduced, allowing for easy end opening without requiring a separate tool. The stay-on-tab design, commonly used today, was introduced in the 1970s and allows the tab to remain attached to the end after opening.

Aluminum beverage can end production is a widespread industry, with manufacturing facilities located in various countries with diverse climates. As a result, the production line operates at different ambient air temperatures, potentially impacting the entire manufacturing process and final material properties. Ambient air temperature plays an important role in an industrial environment and, in extreme cases, can exceed 40 °C in the summer season. For example, a lower material temperature may lead to an increase in the material’s deformation force, and exhibit lower formability and marked springback [5]. The aluminum alloys indicate limited cold formability [6]. Many studies have shown that the forming processes can be improved via the hot stamping process [7,8]. The mechanical properties of aluminum alloys significantly impact the forming process during production. This has been shown both experimentally and via FEM simulation of the can-forming process [9]. The finite element method has been applied to failure prediction of warm deep drawing of cylindrical cups [10]. The literature has extensively investigated the stamping technology applied, especially to steel sheets [11,12]. Pereiraet et al. [13] investigated friction and deformation-induced heating during stamping under room temperature conditions. The model of a stamping process showed that the heat generated during production conditions can result in high temperatures of up to 108 °C in the material, respectively, for what were cold-forming conditions. Tröberet et al. [14] investigated the influence of process parameters on the temperature development in the forming zone. Their study showed that a temperature rise occurs between the tool and workpiece during the forming process due to frictional heating, which affects the whole forming process and tool wear. Yamazaki et al. [15] investigated a new tooling system for forming aluminum beverage can ends using three-dimensional finite element models. Their study showed that thinning of the shell end formed via the proposed tooling system improved by about 3.6%. However, once the end has been produced, it becomes crucial to evaluate its mechanical properties under the internal pressure of the contents in different climatic conditions. Therefore, understanding its mechanical properties under internal pressure from the contents after end production is crucial. The properties of the material of the ends are affected by the aging process of aluminum alloys. Aging processes in aluminum alloys occur at different rates based on the temperature and climatic conditions of both production and storage, ultimately affecting the final mechanical properties of the end.

The technology behind easy-open ends has remained largely unchanged over the years. Recent decades have seen a trend towards reducing end diameter and metal gauge to reduce costs [15,16] and minimize environmental impact [17]. To achieve this, new lightweight ends have been developed, which have a different shape from the long-standing industry standard B64 end developed by Alcoa. The modified geometry of these lightweight ends allows for the use of lighter gauge metal while maintaining the same strength and resistance to buckling under internal pressure within the can. Most ends are manufactured to withstand a minimum internal pressure of 620 kPa (90 psi) [18].

The pressure inside a container depends on the type of drink and temperature. Cans of carbonated soft drinks contain carbon dioxide under pressure. During manufacturing, the gas is solubilized in the liquid at high pressure and low temperature. However, once shipped and stored, the pressure in the headspace increases as the carbon dioxide leaves the solution and enters the headspace due to decreasing pressure. Increasing temperature also decreases solubility, and thus, a can of soda at 20 °C has an internal pressure of about 250 kPa, whereas, at 60 °C inside a hot car, the pressure can increase to 640 kPa due to gas expansion. This high internal pressure strains the end of the can, inducing tensile stresses and can potentially lead to buckling. Moreover, beverages’ can ends show a significant time-dependent decrease in buckle strength, which is important to consider not only in terms of the expiration date of the content, but also the packaging itself. Studies have shown that the production date of the packaging affects changes in mechanical properties [19].

As indicated in the review above, changes in internal pressure occur during the storage, transportation, and heat treatment of beverage cans. In addition, the mechanical properties of the can material itself change over time and with temperature fluctuations, which are influenced by the aging processes in aluminum alloys and their temperature dependence. Therefore, accurately predicting the maximum internal pressure a can withstands under various conditions, such as storage time, temperature, and can material temperature, remains a pressing practical challenge that has yet to be fully resolved.

A literature review reveals that several reports have examined aluminum alloy sheet processing principles and thermal treatment temperatures [20,21]. Yang et al. [22] investigated the effects of bake softening and precipitation behaviors on AA5182 H19 sheets, concluding that the temperature and time of coating processes affect the alloy strength and hardness. Picu et al. [23] studied the mechanical behavior of AA5182 at temperatures ranging from −120 °C to 150 °C, finding that dynamic strain aging impacts ductility and strain hardening. Reid et al. [24] analyzed the buckling of a can with a deformed sidewall, but did not investigate the buckling of an end. However, a comprehensive literature review suggests a gap in studies evaluating the material properties of beverage ends made of AA5182, particularly concerning their lifetime, during which the product undergoes various processes that affect its buckling resistance. None of these previous studies have considered the influence of climatic temperature conditions on the production and operation of beverage can ends, which is an essential practical concern. Consequently, the primary objective of this study is to simulate the production process under various temperature conditions using the finite element method (FEM) and conduct experimental investigations to establish the relationship between the strength of beverage can ends made from alloy AA5182 and the duration and temperature of storage. By addressing this significant research gap, our study aims to provide valuable insights into the influence of climatic temperature conditions on the production and mechanical behavior of beverage can ends. This deeper understanding of their performance will aid in making informed decisions regarding storage conditions and optimizing the operation of beverage cans. Ultimately, our findings will contribute to ensuring the safety and quality of these products in different climatic zones, benefiting the can-making industry and consumers alike.

## 2. Materials and Methods

The H48 temper of the aluminum alloy AA5182 is widely used in the production of beverage can ends due to its favorable material characteristics, such as low density, high corrosion resistance, and excellent formability, enabling the fabrication of intricate shapes. These aluminum alloys have the highest strength among non-heat treatable alloys. The mechanical properties of non-heat treatable alloys are lost if subsequent heating is performed on the cold worked alloy [25]. Table 1 provides the chemical composition of the AA5182 base materials used in this study [26].

The manufacturing process of AA5182 can end stock involves several steps, including smelting, rolling, and coating processes carried out in the final stage. The baking time and temperature of the coating processes depend on the coating line and type of coatings used. Before baking, AA5182 coils are H19 tempered, but after baking, they are in the recovered stage H48, which results in a decrease in strength and hardness and an increase in elongation [22]. The surface of can end stock is covered with polymeric coatings [27] on both sides of the coil, serving the dual purpose of protecting the final product from corrosion and acting as a lubricant during the stamping processes in the manufacturing process [28].

In general, the can-making industry uses two primary standards: the traditional B64 end and the newer lightweight CDL beverage end. This study focused on investigating the cross-section of a 202B64 end, as shown in Figure 1. The manufacturing process of the ends involves cold-forming sheets with a thickness of 0.224 mm [29].

To investigate the relationship between stress and strain at different temperatures, a Zwick/Roell-Z250 machine (ZwickRoell GmbH & Co. KG, Ulm, Germany) equipped with a furnace for heating the samples was used. The test samples were modified to provide tensile tests in the temperature chambers. For this purpose, holes were made at both ends of the sample and the use of flat fixing clamps allowed us to perform a tensile test in the environment of set temperatures. The modified tensile test specimens, as shown in Figure 2, were cut along the rolling direction. The deformation unit of the Zwick/Roell-Z250 machine and examples of samples after the test are illustrated in Figure 3.

Using the Qform v.10 program (https://www.qform3d.com), two categories of finite element method (FEM) computations were executed. The initial category involved simulating the metal deformation process during the end formation of the can. Subsequently, the second category encompassed the simulation of buckling strength tests, assessing the mechanical stability of the can’s end under internal pressure conditions.

In the first scenario, the calculations aimed to gauge the impact of ambient temperature on the punching force. In the FEM simulation of end forming, the tools shown in Figure 4 were employed. The simulation used a two-dimensional axisymmetric model of an elastic–plastic material. The kinematic parameters governing the tool movement and the geometric conditions were carefully chosen to align with the current manufacturing process at CANPACK (Kraków, Poland). To discretize the volume of the workpiece, we employed 6 nodal triangular finite elements of the second order. With each computational finite element time step, the mesh was reconstructed, adapting to both the workpiece’s geometry and the distribution of strain rate. The friction at the interface between the workpiece and the tool was specified using a combined friction law. This law incorporates not only the empirical friction factor, but also factors such as the normal pressure at the contact and the flow stress value of the deformed material at the contact point. The friction factor’s value, 0.15, was selected from the Qform database for conditions involving cold forging of aluminum alloys with mineral oil as a lubricant. In this context, a coupled problem was addressed, wherein the temperature distribution calculated at the ongoing time step influenced the mechanical problem through the modification of the flow stress of the deformed material.

The second scenario of calculations was aimed at determining the amount of internal pressure that would lead to the loss of stability of the can end (buckling strength tests). This calculation was necessary to evaluate the effect of temperature on the value of the critical internal pressure, leading to the loss of stability of the end. With the help of existing equipment, the solution of this problem is experimentally impossible. This problem was solved as a three-dimensional one; the type of element used was a tetrahedron with a linear interpolation of the metal velocities. The internal pressure varied from zero to a value at which the product lost its mechanical stability. In all computational categories for the simulation of large elastic–plastic deformations, Prandtl–Reuss equations were used [31]. Solution convergence was evaluated through rigorous parameters. In the context of the mechanical boundary problem, meticulous analysis was conducted on the flow velocity and average stress at discrete computational steps. For the heat flow problem, scrutiny was directed towards the temperature distribution. These convergence assessments held pivotal significance in guaranteeing the precision and dependability of the solution.

The can ends underwent maximum internal pressure testing after the manufacturing process, using the Machine Versatile Technology LD136B. The test scheme is shown in Figure 5a. The tests were conducted immediately after production and after artificial aging at temperatures ranging from 4 °C to 40 °C, representing potential changes in climatic storage conditions for the ends. The maximum aging time was 4 weeks. Examples of can ends before (Figure 5b) and after destruction due to internal pressure are shown in Figure 5c.

Conclusively, the comprehensive research methodology is visually delineated in Figure 6. According to this scheme, to achieve the study’s objectives, we initiated the sequence of material tests termed “Material study”:-Stress–strain curves were obtained across varying temperatures. These curves were subsequently employed in the segments “FEM simulation of stamping in different temperatures” and “FEM simulation of can end buckling in different temperatures”.-Tensile tests were conducted on the material after aging, aiming to decipher the mechanism underlying the sensitivity of buckle strength to experimental yield stress values, specifically explored in the segments “Tensile tests of material after aging in different temperatures”.

Finally, the influence of aging on buckle strength loss was tested experimentally using special equipment in “Experimental investigation of can ends buckle strength loss”. Using FEM simulation, we obtained deformation force and work over time in different climates (part “Deformation force and work over time in different climates”) and dependence of the buckle strength of the can ends on temperature (part “Dependence of the buckle strength of the can ends on temperature”).

## 3. Results

### 3.1. Material Model

The results of tensile tests were used to obtain a model of the deformable material. The temperature range of the samples was from 20 °C to 150 °C. Examples of stress–strain curves are shown in Figure 7. The approximation of the hardening curves was performed using the following empirical equation:(1)$$\sigma =A\varepsilon^{m_{1}}\exp\left(\frac{m_{2}}{\varepsilon }\right){\left(1+\varepsilon \right)}^{m_{3}}\exp\left(m_{4}\varepsilon \right)\exp\left(-m_{5}t\right)$$
where *ε*—strain, *t*—temperature, *A*, and *m*_1_–*m*_5_—empirical coefficients were determined via the last square method (*A* = 404.8193258; *m*_1_ = 0.009518276; *m*_2_ = −8.18217 × 10^−5^; *m*_3_ = 179.3293196; *m*_4_ = −173.905454; *m*_5_ = 0.001022499).

Figure 7 shows the calculated curves (dashed lines) together with the experimental results (solid lines). The obtained model was implemented in the FEM program, Qform. The irregular behavior in the stress–strain curves at temperatures of 20 °C and 85 °C was the serrated yielding phenomenon, also known as the Portevin–Le Chatelier (PLC) effect. The main factor responsible for the PLC effect is the Mg content in aluminum alloys [32].

### 3.2. Results of FEM Modeling the Forming Process under Different Climatic Conditions

To estimate the temperature boundary conditions for both hot and cold climatic conditions, thermo-vision analysis of deformed tools was conducted at manufacturing plants of the CANPACK company, located in different climatic regions of the world. The temperature distributions on the upper and lower tooling for manufacturing in hot and cold climatic zones are shown in Figure 8 and Figure 9, respectively. Subsequently, two FEM simulations were performed: one for hot climatic conditions (air temperature 40 °C, tool temperature 60 °C) and the other for cold climate (air temperature 20 °C, tool temperature 40 °C). The kinematics of the deformation tools, as shown in Figure 2, were consistent and corresponded to the adopted stamping technology. The temperature distributions obtained for the final state of forming are depicted in Figure 10.

The temperature calculation results showed a difference in metal temperature of about 20 °C. A comparison of the calculated deformation force and work over time, shown in Figure 11, showed that in hot climates, the maximum stamping force was lower by approximately 15%. The work of plastic deformation in a cold climate is approximately 10% greater than the work performed in a hot climate.

### 3.3. Results of Measuring the Buckling Strength after Aging at Various Warehousing Temperatures

The results of measuring the internal pressure leading to the destruction of the end of the can are shown in Figure 12. In these studies, the material was artificially aged at temperatures of 4 °C, 20 °C, and 40 °C for 4 weeks. Every 7 days, 2–4 samples were taken, and tests were carried out according to the scheme shown in Figure 5. The results were brought to a dimensionless form by dividing the measured pressure by the initial pressure leading to the destruction of the can end immediately after the production process (732.1 kPa). Based on the results of the tests performed, it can be concluded that the relative decrease in the strength of the can end under internal pressure at a temperature of 40 °C is 8% over 4 weeks. At a temperature of 4 °C (which corresponded to storage in refrigerators), this decrease is within the limits of statistical error.

The subsequent phase of the study focused on analyzing the mechanism behind the observed change in the mechanical behavior of the can end. The mechanical behavior of beverage can ends is attributed to the features of the aging process of the cold-formed AA5182 alloy. This alloy experiences a reduction in strength over time following the cold stamping process. This is due to the dissolution of second-phase particles, which would otherwise increase the material’s strength through aging hardening. This process is accelerated with an increase in temperature; however, in temperatures in the range of 4 °C to 40 °C, it is not significant. On the other hand, excessive aging time may result in stress relaxation, leading to a further decrease in strength. Both the dissolution of second-phase particles and stress relaxation are diffusion processes. As is well-known, the rate of diffusion processes is initially high, but decreases with time. This particularly explains why the tested strength in Figure 12 decreases rapidly in the first days of aging (in our case, the first 7 days). To validate the aging mechanism in the AA5182 alloy after cold stamping under the conditions employed in our investigation, we performed similar microscopic experiments using an optical microscope. Figure 12 illustrates the microstructure of the material before aging (top photo) and after aging (bottom photo).

An interesting conclusion is drawn from an attempt to compare these results with the corresponding measurement of the mechanical properties of material (Figure 13). The observed changes in material strength are within the limits of measurement error; that is, they are not statistically significant. This indicates that the method of measuring strength by applying internal pressure is more sensitive to material aging and therefore more practical. The tensile test showed no significant changes between the tested sheets of aluminum alloy. In theory, the sheet (CES) should not be susceptible to aging because the after-coating was annealed by the producer of the coil.

## 4. Discussion

The results of the FEM simulation of the stamping process indicate that an increase in ambient temperature from 20 to 40 °C leads to an average reduction in deformation force of about 15% during the product forming stage. However, this difference is more pronounced in the initial stages of forming, becoming considerably less significant in the final stages of stamping (Figure 11a). This effect can be attributed to the increasing proportion of axial compression of the metal in the final stages of forming, which is less sensitive to temperature variations compared to shape change. The corresponding difference in the magnitude of deformation work amounts to approximately 10% (Figure 11b). Overall, these findings have practical implications for predicting energy consumption during production.

The impact of temperature–time conditions on the mechanical behavior of the finished product was experimentally investigated using a specialized testing setup, as shown in Figure 5. The investigations revealed that in hot climates (at a temperature of 40 °C), the finished product experiences an 8% reduction in buckle strength after one month of storage. At a storage temperature of 20 °C, this reduction amounts to 6%, while no changes in properties were observed when stored in a refrigerator (at a temperature of 4 °C). These results hold significant practical importance.

An interesting observation is that the changes in the material’s strength characteristics after corresponding aging at different temperatures are not as noticeable (Figure 13). This implies that the applied strength test of the finished product through internal pressure loading is much more sensitive to changes in material properties due to aging. Furthermore, it appears that the influence of aging on the material cannot be solely described by the ultimate strength characteristics obtained from standard tensile tests, but rather requires a more detailed comparison of stress–strain curves.

Among the unresolved questions in this paper, the direct influence of material temperature on the buckle strength of can ends stands out. The test shown in Figure 5 cannot be conducted at elevated temperatures. Moreover, using Equation (1) directly to estimate the relative change in the ultimate strength of the can end under pressure is not feasible for two reasons. Firstly, there is a weak correlation observed and described earlier between the results of the tensile test and the strength of the can end under pressure. Secondly, the mechanism of can end strength loss under pressure involves a rapid loss of mechanical stability, making it a highly nonlinear process. For these reasons, the relationship between the buckle strength of the can end and the material temperature was obtained through numerical calculations. The Qform v10.1 software was used for modeling, employing a three-dimensional approach with boundary conditions set according to the scheme in Figure 5. The choice of this software was justified by its previous usage in modeling plastic deformation stability loss processes, as demonstrated in reference [33]. An elastic–plastic material model was utilized, and stress–strain curves were directly obtained from experimental results for different temperatures (examples of such curves are shown in Figure 6).

During the simulation, internal pressure was gradually increased from 0 kPa until the point of instability in the system. An example calculation for a temperature of 20 °C is presented in Table 2. According to these data, at a pressure of 740 kPa, the material’s velocity sharply increases in the direction of pressure application, indicating the loss of stability in the can end.

The mechanism of stability loss is further illustrated in Figure 14, displaying the results for a material temperature of 85 °C. As is evident from this figure, the instability process occurs almost instantaneously, facilitating the determination of the critical pressure from the calculation results. Another observation from the simulation results reveals the asymmetric nature of the can end deformation, which corresponds to the observed material shape in the experiment (Figure 5c).

The summary of the calculations, in the form of a critical pressure–temperature dependency, is presented in Figure 15. For a temperature of 20 °C, a comparison between the simulation and experimental results is possible (shown in Figure 15), while for other temperatures, the results are based solely on simulations. The obtained results allow us to assess the loss of can end buckle strength under internal pressure at different material temperatures. For instance, at a temperature of 85 °C, the can end loses 5% of its strength compared to buckle strength at 20 °C. This reduction can be interpreted as significant, considering that as the temperature increases, the internal pressure also rises, leading to a decrease in the buckle strength due to the aging of the alloy over time and a decrease in the current buckle strength under the direct influence of elevated temperature.

## 5. Conclusions

This study aimed to investigate the influence of temperature climatic conditions on the manufacturing process and operation of beverage can ends, with the goal of ensuring the safety and quality of the products. Based on the findings of this study, the following conclusions can be drawn.

The numerical study revealed that temperature conditions can significantly impact the parameters of the forming process, with a remarkable effect on stamping force (15%) and energy of deformation (10%). These findings are important for planning the energy efficiency of the technology.

Experimental investigations showed that artificial aging of the material of can ends at temperatures of 4 °C, 20 °C, and 40 °C for 4 weeks leads to a relative decrease in the buckle strength of the end of the can when loaded with internal pressure by 8% at the aging temperature of 40 °C. This value is significant and should be taken into account during the operation of beverage can ends. The reduction in can end buckle strength after 4 weeks of aging in cold climates (20 °C) is about 6%, while in refrigerator conditions (4 °C), no significant influence of aging on the buckle strength of the end was observed.

FEM simulations of the buckle strength of can ends under pressure showed that this is a non-linear process, influenced by the loss of the stability mechanism. The change in material temperature from 20 °C to 85 °C resulted in a decrease in buckle strength of up to 5%.

Although our focus throughout this paper has been on the influence of temperature on the behavior of beverage ends in production and storage, the potential benefits of our findings are wider. Observed stamping force decrease may lead to some energy saving, and it can be worth the effort, as any calculation of final energy would be suitable to confirm this statement, especially regarding environment implications. These possibilities are yet to be researched. Additionally, future research could investigate the influence of other climatic conditions, such as humidity and atmospheric pressure, on the production and operation of beverage can ends.

## Figures and Tables

**Figure 1 materials-16-06137-f001:**
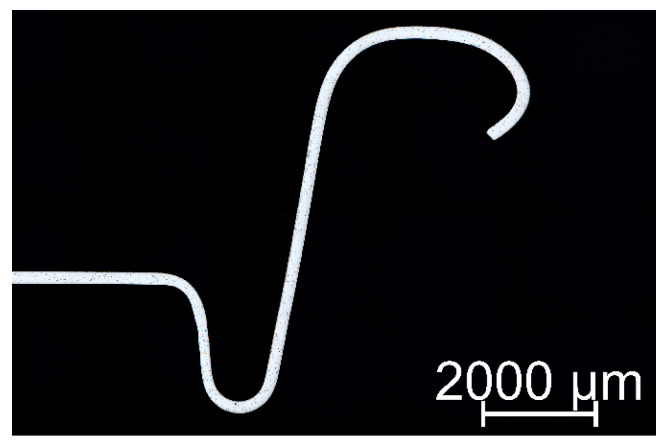
Cross-section through a 202B64 beverage can end.

**Figure 2 materials-16-06137-f002:**
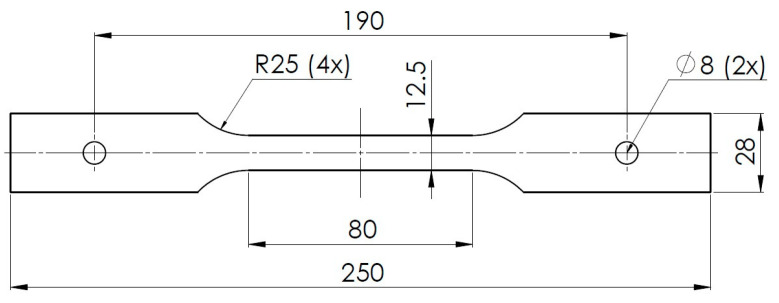
Modified specimen (mm) for the tensile tests based on standard sample (PN-EN 10002-1+AC1) [30].

**Figure 3 materials-16-06137-f003:**
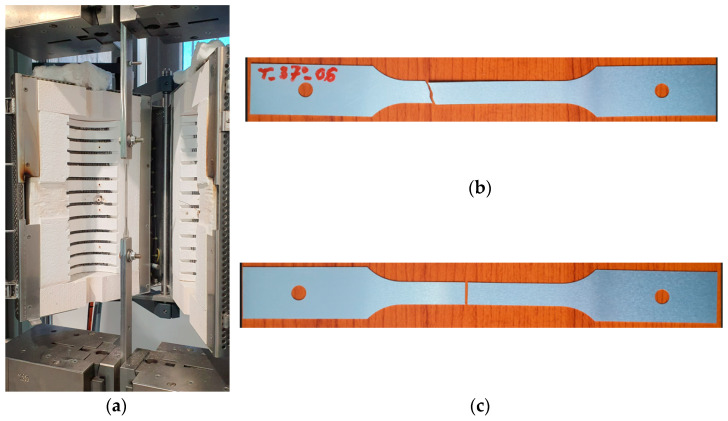
Tensile test of the initial sheet material at different temperatures: (**a**) machine Zwick/Roell-Z250 for tensile tests; (**b**) sample after test at 37 °C; (**c**) sample after test at 81 °C.

**Figure 4 materials-16-06137-f004:**
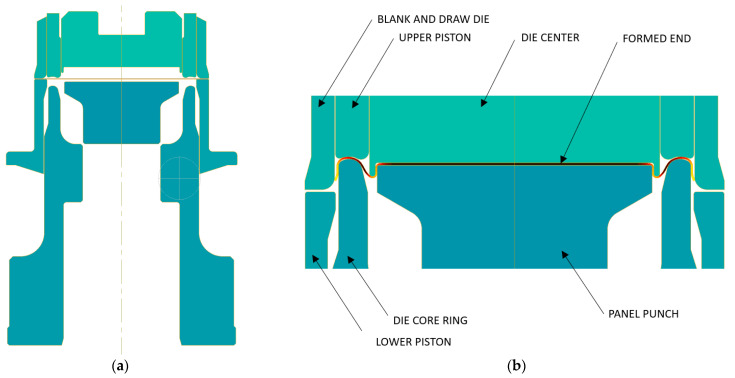
Geometrical conditions of the shell-forming process: (**a**) initial positions of tools and blank in 2-D draw; (**b**) final positions of tools after forming finished.

**Figure 5 materials-16-06137-f005:**
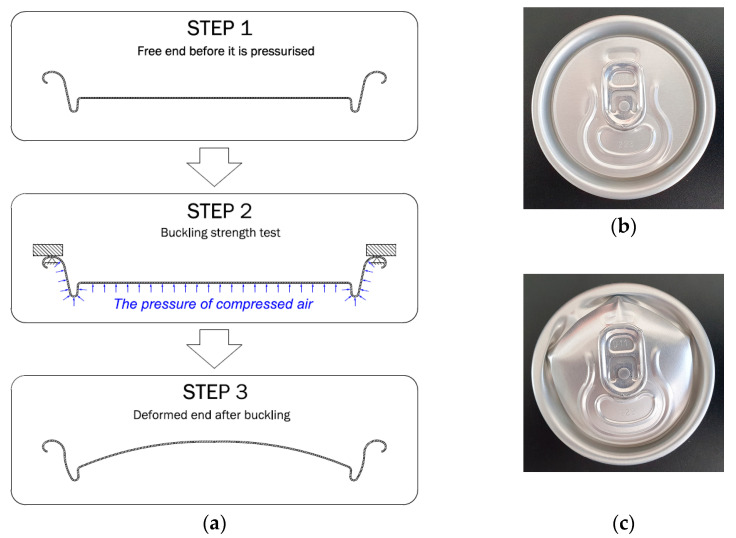
Buckling strength testing: (**a**) test scheme; (**b**) beverage can end before the buckling test; (**c**) beverage can end after the buckling test.

**Figure 6 materials-16-06137-f006:**
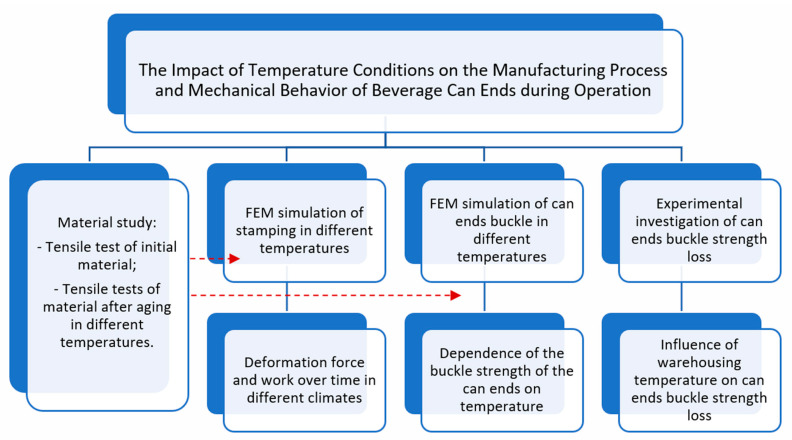
Overall flowchart of the research methodology.

**Figure 7 materials-16-06137-f007:**
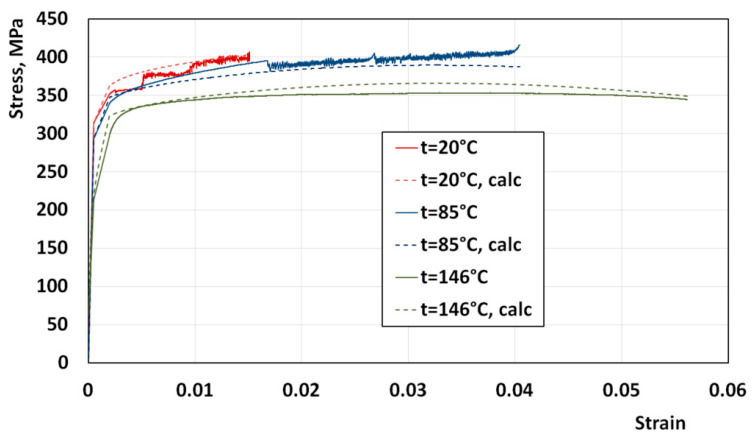
Examples of stress–strain curves during tensile tests of 5182 alloy sheets in different temperatures.

**Figure 8 materials-16-06137-f008:**
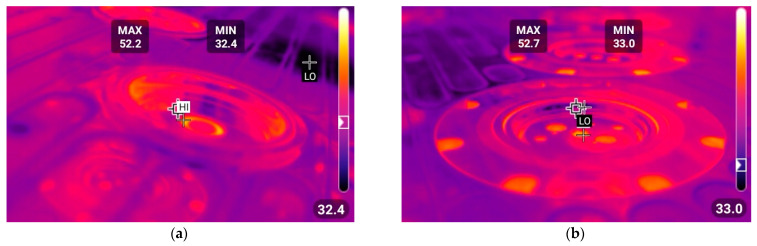
Temperature distributions via the thermal imaging camera for hot climatic conditions at 40 °C: (**a**) upper tooling; (**b**) lower tooling.

**Figure 9 materials-16-06137-f009:**
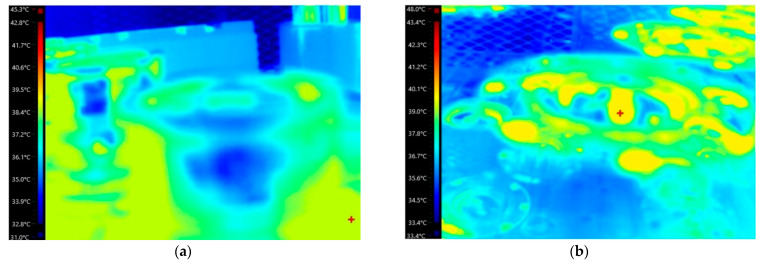
Temperature distributions via the thermal imaging camera for cold climate at 20 °C: (**a**) upper tooling; (**b**) lower tooling.

**Figure 10 materials-16-06137-f010:**
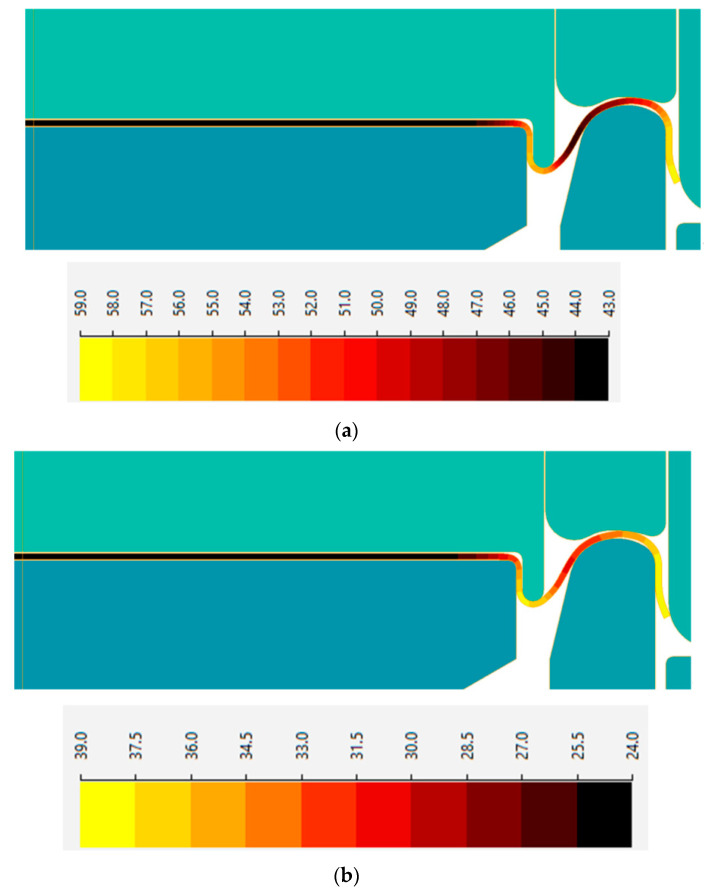
Temperature distributions: (**a**) hot climatic conditions at 40 °C; (**b**) cold climate at 20 °C.

**Figure 11 materials-16-06137-f011:**
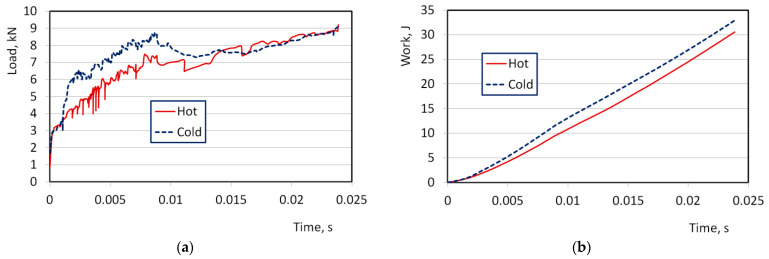
Comparison of the calculated deformation load (**a**) and work (**b**) over time.

**Figure 12 materials-16-06137-f012:**
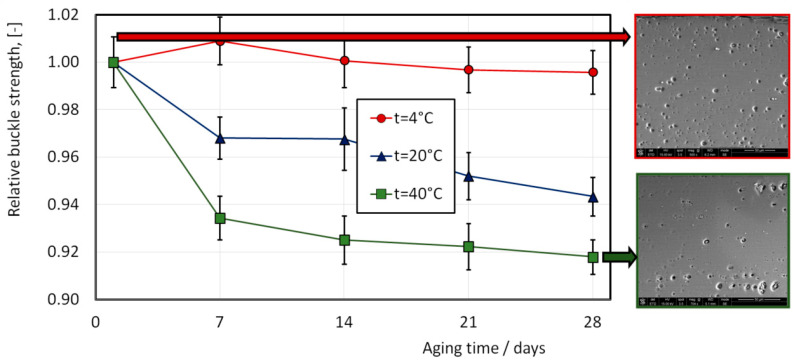
The influence of warehousing temperature on can ends’ buckle strength loss and microstructure (buckle strength before aging 732.1 kPa).

**Figure 13 materials-16-06137-f013:**
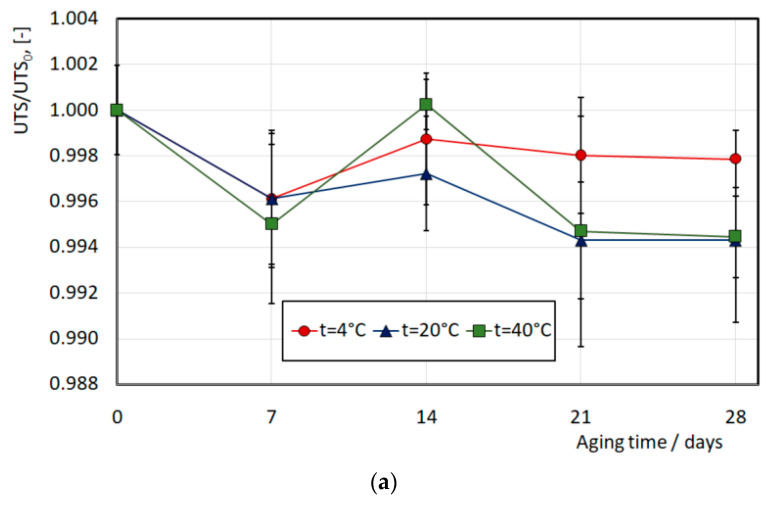
The influence of warehousing temperature on UTS (**a**) and YS (**b**), (UTS_0_ = 421.0 MPa, YS_0_ = 349.9 MPa).

**Figure 14 materials-16-06137-f014:**
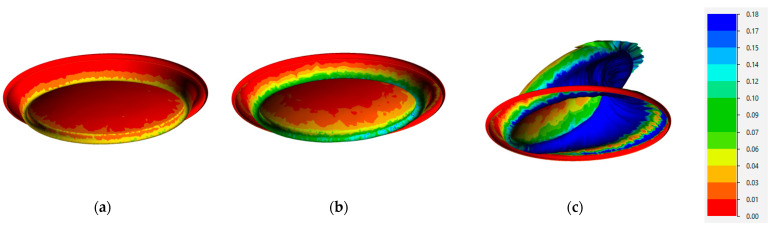
Distribution of strain intensity during buckling of the can end at 85 °C under internal pressure (**a**) 100 kPa, (**b**) 700 kPa, (**c**) 700.1 kPa.

**Figure 15 materials-16-06137-f015:**
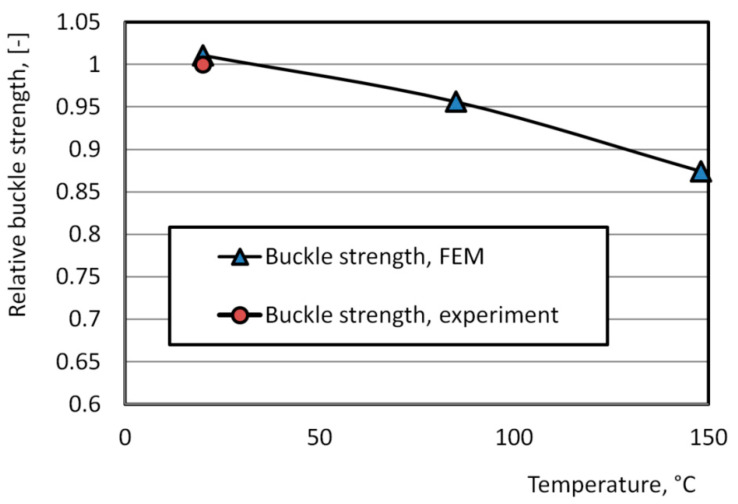
Dependence of the relative buckle strength of the can ends on temperature (buckle strength 732.1 kPa corresponded to relative buckle strength 1.0).

**Table 1 materials-16-06137-t001:** Nominal chemical composition (wt%) of the AA5182 alloy [26].

Grade Designation	Si	Fe	Cu	Mn	Mg	Cr	Zn	Ti	Unspecified Other Elements		Al
									Each	Total	
5182	0.20	0.35	0.15	0.20–0.50	4.0–5.0	0.10	0.25	0.1	0.05	0.15	Remainder

**Table 2 materials-16-06137-t002:** Results of FEM simulation of buckling of can ends at room temperature.

Distribution of Velocity in Vertical Direction, mm/s	P, kPa
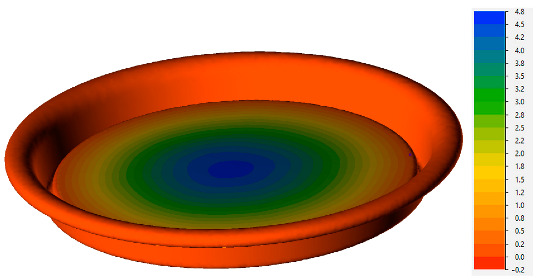	100
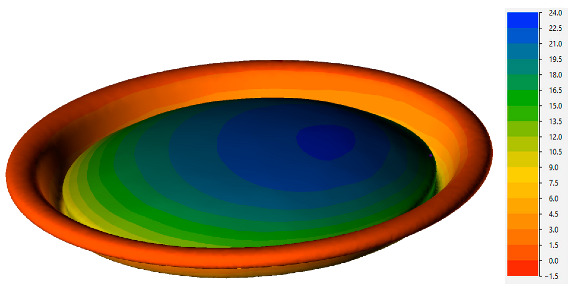	680
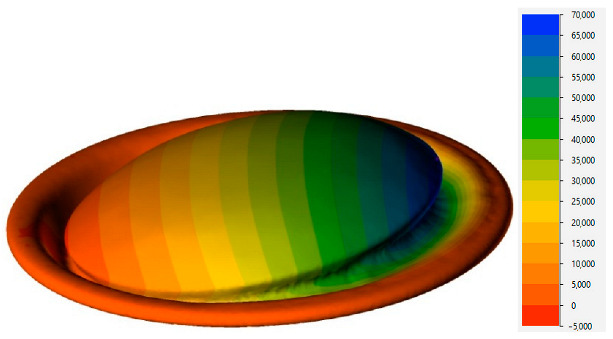	740

## Data Availability

Data are provided within the article.

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
