# Peer review of "The Impact of Temperature Conditions on the Manufacturing Process and Mechanical Behavior of Beverage Can Ends during Operation"

_materials, 2023, doi:10.3390/ma16186137_

Round 1

Reviewer 1 Report

the current manuscript is studying the effect of temperature rise in some hot countries on the forming process of the beverage cans, honestly I like the idea but the soundness of the process is not applicable and I am forced to reject the manuscript for the following reasons:

1) normally the factories use air conditions in hot climatic countries to protect the life span of the forming machines, thus the scope of the study is not significant.

2) there is no microstructure analysis to confirm the claims of the results mentioned by the authors

3) the design process by solidworks is not accurate for a research article, I recommend using Ansys for proper Finite element analysis.

for these reasons I recommend a rejection of the paper in the Materials Jounal. 

the english is average level and major corrections are needed

Author Response

Point-by-Point Response to Reviewer

Dear Reviewer,
Thank you very much for your time and all your effort in evaluating our manuscript. We have incorporated your valuable suggestions and have made corrections as marked in the revised manuscript, which we hope will meet with your approval. Please find our reply below. 

Response to Reviewer #1 Comments 

Point 1: normally the factories use air conditions in hot climatic countries to protect the life span of the forming machines, thus the scope of the study is not significant.

Response 1: Thank you for your comments. In general can making industry does not use air conditioners in the production area even in hot climate countries. Perhaps other industries use air conditions to protect the life span of the forming machines. For example, for 2 countries with hot climates where CANPACK company has beverage ends production, in the middle of summer the ambient air temperature of the production line exceeds 40 °C. For research paper purpose, the tests by using the thermal imaging cameras have been performed in 2 locations: Poland (cold climatic conditions) and United Arab Emirates (hot climatic conditions).

The following explanation has been added to the paper: “Ambient air temperature plays an important role in an industrial environment and in extreme cases can exceed 40 °C in the summer season”.

Point 2: there is no microstructure analysis to confirm the claims of the results mentioned by the authors.

Response 2: Thank you for your insightful comments. Changes in the AA5182 alloy occurring at the considered relatively low temperatures 4-40 °C lead to changes in the mechanical behavior of the final product, which was the subject of study in this article. It was quite difficult to register the relevant significant changes in the microstructure. We used an optical microscope and found no significant difference in microstructure during aging at different temperatures. Some pictures showed a decrease in the number of inclusions after aging, but there were no differences in others. It follows from this that more powerful tools should be used to study microstructural changes, which we will probably do in our future studies. However, in this article, we set another task related to the analysis of the influence of temperature on the mechanical behavior of beverage can ends during operation and manufacture. This issue is very important in practice, and its solution will help optimize both the production process and the use of cans in different climates.

Therefore, from our point of view, detailed microstructure analysis was out of the aim of our paper. However, because you asked about it, we added to the paper the following explanation about microstructure:

„The mechanical behavior of beverage can ends is attributed to the features of the aging process of cold-formed AA5182 alloy. This alloy experiences a reduction in strength over time following the cold stamping process. This is due to the dissolution of second-phase particles, which would otherwise increase the material's strength through aging hardening. This process is accelerated with an increase in temperature however, in temperatures in the range 4-40 °C it is not significant. On the other hand, excessive aging time may result in stress relaxation, leading to a further decrease in strength. Both the dissolution of second-phase particles and stress relaxation are diffusion processes. As is well-known, the rate of diffusion processes is high initially but decreases with time. This particularly explains why the tested strength in Figure 12 decreases rapidly in the first days of aging (in our case, the first 7 days)”.

Point 3: the design process by solidworks is not accurate for a research article, I recommend using Ansys for proper Finite element analysis.

Response 3: Thank you for your comments. The SolidWorks program was not used for process design, but only for a qualitative explanation of the phenomenon of asymmetry of the loss of stability. The entire analysis was performed using the Qform program, which was originally designed to analyze the processes of large elastic-plastic deformations (Figure 14, Table 2). Apparently, the use of Solidworks to confirm the asymmetry obtained using Qform and in the experiment introduces unnecessary confusion, so we decided to remove these results from the paper.

NOTE: All references in the text refer to figure numbers for the modified manuscript.

Thank you and best regards.

Yours sincerely,

Paweł Kokoszka

CANPACK S.A., Business Support Service, Starowiejska 28 Street, 32-800 Brzesko, Poland

E-mail: pawel.kokoszka@canpack.com

Reviewer 2 Report

Dear Authors,

My comments are:

1. Title needs revision, lacks specificity on temperature conditions, process, and mechanical behavior studied. OPTIONAL

2. FEM model and boundary conditions inadequately explained. This is not how you perform FEA, where is convergence study, mesh info etc?

3. Sole reliance on 2D analysis raises accuracy concerns for real-world representation. The coupled results are nowhere for 3D

 4. Methodology and problem statement lack coherence and logical flow.

 5. Mathematical writing requires professional revision. Use relevant software. 

6. Results section needs clearer explanations, focused on relevant phenomena.

 7. Absence of literature and model comparison.

 8. Lack of details on model validation against specific boundary conditions. There are multiple cases if we talk realistic.

 9. Unclear scope of model validation – broader conditions or specifics?

10. Overall structure improvements needed: refined title, organized methodology, enhanced math presentation, clearer results with comparisons, and validation clarification.

Moderate

Author Response

Point-by-Point Response to Reviewer

Dear Reviewer,
Thank you very much for your time and all your effort in evaluating our manuscript. We have incorporated your valuable suggestions and have made corrections as marked in the revised manuscript, which we hope will meet with your approval. Please find our reply below. 

Response to Reviewer #2 Comments

Point 1: Title needs revision, lacks specificity on temperature conditions, process, and mechanical behavior studied. OPTIONAL

Response 1: We have slightly modified title to make it more concise. The new title:

“The Impact of Temperature Conditions on the Manufacturing Process and Mechanical Behavior of Beverage Can Ends during Operation”.

Point 2: FEM model and boundary conditions inadequately explained. This is not how you perform FEA, where is convergence study, mesh info etc?

Response 2: Thank you for your comments. The following description has been incorporated into the manuscript:

“Using the Qform v.10 program (https://www.qform3d.com), two categories of Finite Element Method (FEM) computations were executed. The initial category involved simulating the metal deformation process during the end formation of the can. Subsequently, the second category encompassed the simulation of buckling strength tests, assessing the mechanical stability of the can's end under internal pressure conditions.

In the first scenario, the calculations aimed to gauge the impact of ambient temperature on the punching force. In the FEM simulation of end forming, the tools shown in Figure 4 were employed. The simulation used a two-dimensional axisymmetric model of an elastic-plastic material. The kinematic parameters governing the tool movement and the geometric conditions were carefully chosen to align with the current manufacturing process at CANPACK. To discretize the volume of the workpiece, we employed 6 -nodal triangular finite elements of the second order. With each computational Finite Element time step, the mesh was reconstructed, adapting to both the workpiece's geometry and the distribution of strain rate. The friction at the interface between the workpiece and the tool was specified using a combined friction law. This law incorporates not only the empirical friction factor but also factors such as the normal pressure at the contact and the flow stress value of the deformed material at the contact point. The friction factor's value, 0.15, was selected from the Qform database for conditions involving cold forging of aluminum alloys with mineral oil as a lubricant. In this context, a coupled problem was addressed, wherein the temperature distribution calculated at the ongoing time step influenced the mechanical problem through the modification of the flow stress of the deformed material.

The second scenario of calculations was aimed at determining the amount of internal pressure that would lead to the loss of stability of the can end (buckling strength tests). This calculation was necessary to evaluate the effect of temperature on the value of the critical internal pressure, leading to the loss of stability of the end. With the help of existing equipment, the solution of this problem experimentally is impossible. This problem was solved as a three-dimensional one, the type of elements used was a tetrahedron with a linear interpolation of the metal velocities. The internal pressure varied from zero to a value at which the product lost its mechanical stability.

In all computational categories for the simulation of large elastic-plastic deformations, Prandtl-Reuss equations were used [30]. Solution convergence was evaluated through rigorous parameters. In the context of the mechanical boundary problem, meticulous analysis was conducted on the flow velocity and average stress at discrete computational steps. For the heat flow problem, scrutiny was directed towards the temperature distribution. These convergence assessments held pivotal significance in guaranteeing the precision and dependability of the solution".

Point 3: Sole reliance on 2D analysis raises accuracy concerns for real-world representation. The coupled results are nowhere for 3D

Response 3: This aspect has been previously addressed to some extent (see Response 2, please). The application of 2D simulation was constrained to the analysis of the stamping process, which inherently possesses a high degree of axisymmetric accuracy. For all other scenarios, the analysis was conducted within the realm of three-dimensional problems.

Point 4: Methodology and problem statement lack coherence and logical flow.

Response 4: Thank you for your valuable time and insightful comments. We have tried to improve the description of the methodology by adding an Overall flowchart of the research methodology. The following description has been incorporated into the manuscript:

“Conclusively, the comprehensive research methodology is visually delineated in Figure 6. According to this scheme, to achieve the study's objectives, we initiated the sequence of material tests termed "Material study":

  • Stress-strain curves were obtained across varying temperatures. These curves were subsequently employed in the segments "FEM simulation of stamping in different temperatures" and "FEM simulation of can end buckling in different temperatures".
  • Tensile tests were conducted on the material after aging, aiming to decipher the mechanism underlying the sensitivity of buckle strength to experimental yield stress values, specifically explored in the segments "Tensile tests of material after aging in different temperatures".

Finally, the influence of aging on buckle strength loss was tested experimentally using special equipment in part “Experimental investigation of can ends buckle strength loss”. 

By FEM simulation we obtained deformation force and work over time in different climates (part "Deformation force and work over time in different climates") and dependence of the buckle strength of the can ends on temperature (part "Dependence of the buckle strength of the can ends on temperature").

Point 5: Mathematical writing requires professional revision. Use relevant software. 

Response 5: Agree, we have made some revisions to the mathematical writing, sincerely hoping it will meet your requirement.

Point 6: Results section needs clearer explanations, focused on relevant phenomena.

Response 6: Thank you for your insightful comments. The following explanation has been added to the paper:

“The irregular behavior in the stress-strain curves at temperatures, 20 °C and 85 °C, was the serrated yielding phenomenon, also known as the Portevin-Le Chatelier (PLC) effect. The main factor responsible for the PLC effect is the Mg content in aluminum alloys [31]”.

And additionally added:

„The mechanical behavior of beverage can ends is attributed to the features of the aging process of cold-formed AA5182 alloy. This alloy experiences a reduction in strength over time following the cold stamping process. This is due to the dissolution of second-phase particles, which would otherwise increase the material's strength through aging hardening. This process is accelerated with an increase in temperature however, in temperatures in the range 4-40 °C it is not significant. On the other hand, excessive aging time may result in stress relaxation, leading to a further decrease in strength. Both the dissolution of second-phase particles and stress relaxation are diffusion processes. As is well-known, the rate of diffusion processes is high initially but decreases with time. This particularly explains why the tested strength in Figure 12 decreases rapidly in the first days of aging (in our case, the first 7 days)”.

Point 7: Absence of literature and model comparison.

Response 7: Thank you for your insightful comments. The following explanation has been added to the paper:

“Yamazaki et al. [15] investigated a new tooling system for forming aluminum beverage can ends use three-dimensional finite element models. Their study showed that thinning of the shell end formed by the proposed tooling system has been improved about 3.6%”.

And additionally added:

"The second scenario of calculations was aimed at determining the amount of internal pressure that would lead to the loss of stability of the can end (buckling strength tests). This calculation was necessary to evaluate the effect of temperature on the value of the critical internal pressure, leading to the loss of stability of the end. With the help of existing equipment, the solution of this problem experimentally is impossible. This problem was solved as a three-dimensional one, the type of elements used was a tetrahedron with a linear interpolation of the metal velocities. The internal pressure varied from zero to a value at which the product lost its mechanical stability.

In all computational categories for the simulation of large elastic-plastic deformations, Prandtl-Reuss equations were used [30]. Solution convergence was evaluated through rigorous parameters. In the context of the mechanical boundary problem, meticulous analysis was conducted on the flow velocity and average stress at discrete computational steps. For the heat flow problem, scrutiny was directed towards the temperature distribution. These convergence assessments held pivotal significance in guaranteeing the precision and dependability of the solution”.

Point 8: Lack of details on model validation against specific boundary conditions. There are multiple cases if we talk realistic.

Response 8: Thank you for your valuable time and insightful comments. To validate the model of can ends buckle at different temperatures, we used experimental data on room temperature buckle strength obtained on the LD136B machine. We incorporated into Qform program the material properties acquired from our tensile tests.  The boundary conditions were set strictly following the test scheme - fixing the displacement at the edge of the sample and normal pressure under Figure 5. During the calculation, the pressure increased from 0 to a value corresponding to the loss of stability. In Table 2, this value is given (740 kPa). This pressure value was compared with the experimental data obtained on a testing machine according to the scheme in Figure 5 (732.1 kPa, Figure 12). Although the validation was carried out for one value, it should be emphasized that close values in the calculation and experiment indicate that all model components, such as grid parameters, boundary conditions, time step, material properties, and others, are selected correctly. On the other hand, obtaining experimental data on the used LD136B testing machine for elevated temperatures is technically impossible. It was the receipt of such information for different temperatures that was the motivation for performing the FEM calculation.

Point 9: Unclear scope of model validation – broader conditions or specifics?

Response 9: Thank you for your comments. The specific conditions for performing the buckle strength test (on the Machine Versatile Technology LD136B it can only be performed at room temperature) made it possible to perform only one comparison of the calculation and experiment - at 20 °C.

Point 10:  Overall structure improvements needed: refined title, organized methodology, enhanced math presentation, clearer results with comparisons, and validation clarification

Response 10: Thank you for your comments. We tried to do it. Since this remark coincides with one of the remarks (point 4) I take the liberty of referring you to the answer above.

NOTE: All references in the response refer to figure numbers and references numbers for the modified manuscript.

With kind regards,

Yours sincerely,

Paweł Kokoszka

CANPACK S.A., Business Support Service, Starowiejska 28 Street, 32-800 Brzesko, Poland

E-mail: pawel.kokoszka@canpack.com

Reviewer 3 Report

In this study, the authors investigated the influence of temperature climatic conditions on the manufacturing and operation of Al5182 can ends. The authors used both experimental and simulation results to explain how the climatic conditions affect the related performance. The results are interesting. However, from the prospect of material side, the related discussion is not satisfied and need to be improved. Here are some points to be modified.

1.       At line 233 and 234, the author mentioned the heat treatment can reduce the inclusions during aging process, according to ref 28. Firstly, there are no such conclusions at Ref 28. In their study, the annealing heat treatment reduced the amount of secondary phases, not inclusions. Secondary, the secondary phase can be dissolved into Al matrix, only at high temperature (over 250C), not at such low temperature the authors used (20C and40C). It is impossible to dissolve secondary phases at such low temperature, from thermodynamic prospects. For inclusions in Al alloys ( such as Al2O3), the dissolve process are more difficult. Please re-write this part about how the aging process affect the related mechanical properties.

2.       The  SEM images show the differences of inclusion amount. It looks more likely sample related. The inclusions looks like the formed/etched pores during samples preparation. Meanwhile, if the author indicates they are inclusions, it is suggested to add the EDX results of these points.

3.       For the mechanical properties, the differences between the aged samples are within the accepted error region. It doesn’t seem to have an obvious effect on the mechanical strength at such a low aging temperature. Some Al alloys have natural aging when they are placed at Room temperature, due to the formation of Mg2Si and related precipitates. This natural aging will lead to an increase of YS/hardness, rather than a decrease. The used composition 5182 with low content of Si means the tendency of form Mg2Si precipitates is also low. Therefore, it seems the differences are more just tensile results fluctuation.

Author Response

Point-by-Point Response to Reviewer

Dear Reviewer,
Thank you very much for your time and all your effort in evaluating our manuscript. We have incorporated your valuable suggestions and have made corrections as marked in the revised manuscript, which we hope will meet with your approval. Please find our reply below. 

Response to Reviewer #3 Comments

Point 1: At line 233 and 234, the author mentioned the heat treatment can reduce the inclusions during aging process, according to ref 28. Firstly, there are no such conclusions at Ref 28. In their study, the annealing heat treatment reduced the amount of secondary phases, not inclusions. Secondary, the secondary phase can be dissolved into Al matrix, only at high temperature (over 250C), not at such low temperature the authors used (20C and 40C). It is impossible to dissolve secondary phases at such low temperature, from thermodynamic prospects. For inclusions in Al alloys ( such as Al2O3), the dissolve process are more difficult. Please re-write this part about how the aging process affect the related mechanical properties.

Response 1: Thank you for your valuable time and insightful comments. We have made some revisions of the results section and we have re-written part about how the aging process affect the related mechanical properties. We have added the following information:

“The mechanical behavior of beverage can ends is attributed to the features of the aging process of cold-formed AA5182 alloy. This alloy experiences a reduction in strength over time following the cold stamping process. This is due to the dissolution of second-phase particles, which would otherwise increase the material's strength through aging hardening. This process is accelerated with an increase in temperature however, in temperatures in the range 4-40 °C it is not significant. On the other hand, excessive aging time may result in stress relaxation, leading to a further decrease in strength. Both the dissolution of second-phase particles and stress relaxation are diffusion processes. As is well-known, the rate of diffusion processes is high initially but decreases with time. This particularly explains why the tested strength in Figure 12 decreases rapidly in the first days of aging (in our case, the first 7 days)”.

Point 2: The SEM images show the differences of inclusion amount. It looks more likely sample related. The inclusions looks like the formed/etched pores during samples preparation. Meanwhile, if the author indicates they are inclusions, it is suggested to add the EDX results of these points.

Response 2: Thank you for your comments. Yes, you are most likely right. Indeed, when analyzing microstructures obtained by SEM and using an optical microscope for most samples, we did not see a difference in either the number of inclusions or grain size, and only in a few samples this difference was visible. Apparently, to analyze the correlation between the observed mechanical phenomena and the microstructure, it is necessary to carry out deeper metallographic studies. However, since even a study of yield stress versus aging conditions did not show a statistically significant correlation with aging temperature (although a trend was visible), I doubt that such studies of microstructure will show anything more than what we have now. The only test that showed high sensitivity to the aging of material was the test of loading the finished product with internal pressure.

Thus, given that the goal of our research, which is to study mechanical processes, has been achieved, we consider it possible to carry out in-depth studies in our further studies.

Changes in the AA5182 alloy occurring at the considered relatively low temperatures 4-40 °C lead to changes in the mechanical behavior of the final product, which was the subject of study in this article. It was quite difficult to register the relevant significant changes in the microstructure. We used an optical microscope and found no significant difference in microstructure during aging at different temperatures. Some pictures showed a decrease in the number of inclusions after aging, but there were no differences in others. It follows from this that more powerful tools should be used to study microstructural changes, which we will probably do in our future studies. However, in this article, we set another task related to the analysis of the influence of temperature on the mechanical behavior of beverage can ends during operation and manufacture. This issue is very important in practice, and its solution will help optimize both the production process and the use of cans in different climates.

We have modified the article by writing a more careful interpretation of the microstructure added the same information as Response 1.

Point 3: For the mechanical properties, the differences between the aged samples are within the accepted error region. It doesn’t seem to have an obvious effect on the mechanical strength at such a low aging temperature. Some Al alloys have natural aging when they are placed at Room temperature, due to the formation of Mg2Si and related precipitates. This natural aging will lead to an increase of YS/hardness, rather than a decrease. The used composition 5182 with low content of Si means the tendency of form Mg2Si precipitates is also low. Therefore, it seems the differences are more just tensile results fluctuation.

Response 3: Thank you for your constructive comments. Aluminum alloys that are wrought or mechanically worked can be placed in the two categories of heat treatable and non-heat treatable alloys. To improve the mechanical properties, heat treatable alloys can undergo heat treatment processes such as solid solution strengthening or precipitation hardening (examples of alloy series 6XXX and 7XXX). However, non-heat treatable alloys, rely on deformation in order to develop the required structures and mechanical properties in the alloy. The mechanical properties of non-heat treatable alloys are lost if subsequent heating is performed on the cold worked alloy  (example of alloy series 5XXX).

The following explanation has been added to the paper:

“These aluminum alloys have the highest strength among non-heat treatable alloys. The mechanical properties of non-heat treatable alloys are lost if subsequent heating is performed on the cold worked alloy [25]”.

Additionally, the tensile testing was performed additionally only, because in theory the sheet should not be susceptible to aging. The phenomenon of mechanical strength change for the sheet itself should not matter, because the sheet after coating was annealed by producer of coil. The following explanation provided in the article:

“The manufacturing process of AA5182 can end stock involves several steps, including smelting, rolling and coating processes carried out in the final stage. The baking time and temperature of the coating processes depend on the coating line and type of coatings used. Before baking, AA5182 coils are H19 tempered, but after baking, they are in the recovered stage H48, which results in a decrease in strength and hardness and an increase in elongation [22]”.

and

“An interesting conclusion is drawn from an attempt to compare these results with the corresponding measurement of the mechanical properties of material (Figure 13). The observed changes in material strength are within the limits of measurement error, that is, they are not statistically significant. This indicates that the method of measuring strength by applying internal pressure is more sensitive to material aging and therefore more practical”.

Additionally, the following explanation has been added to the paper:

“The tensile test shown no significant changes between the tested sheets of aluminum alloy. In theory the sheet (CES) should not be susceptible to aging, because after coating was annealed by producer of coil”.

NOTE: All references in the response refer to figure numbers and references numbers for the modified manuscript.

With kind regards,

Yours sincerely,

Paweł Kokoszka

CANPACK S.A., Business Support Service, Starowiejska 28 Street, 32-800 Brzesko, Poland

E-mail: pawel.kokoszka@canpack.com

Reviewer 4 Report

Dear authors,

I congratulate you on the manuscript of the paper, which is useful for industrial application and further scientific research. The problem of the influence of environmental warming in industrial production has not yet been sufficiently investigated and your work is a good contribution to this topic. For a better understanding of the FEM model and possible reproducibility of the simulation, I suggest that you add the following explanations to your paper: - In the Materials and methods chapter, in the description of the FEM simulation, a brief description of the starting mathematical equations, their connections and kinematic parameters of the materials behaviour, that you have chosen in the QForm program; - In conclusion, an idea or guideline should be added for further research on the influence of temperature on the behavior of cans in production and storage.

Author Response

Point-by-Point Response to Reviewer

Dear Reviewer,
Thank you very much for your time and all your effort in evaluating our manuscript. We have incorporated your valuable suggestions and have made corrections as marked in the revised manuscript, which we hope will meet with your approval. Please find our reply below. 

Response to Reviewer #4 Comments

For a better understanding of the FEM model and possible reproducibility of the simulation, I suggest that you add the following explanations to your paper:

Point 1: In the Materials and methods chapter, in the description of the FEM simulation, a brief description of the starting mathematical equations, their connections and kinematic parameters of the materials behaviour, that you have chosen in the QForm program.

Response 1: Thank you for your comments. Of course, we agree with this remark and the corresponding description has been added to the article:

“Using the Qform v.10 program (https://www.qform3d.com), two categories of Finite Element Method (FEM) computations were executed. The initial category involved simulating the metal deformation process during the end formation of the can. Subsequently, the second category encompassed the simulation of buckling strength tests, assessing the mechanical stability of the can's end under internal pressure conditions.

In the first scenario, the calculations aimed to gauge the impact of ambient temperature on the punching force. In the FEM simulation of end forming, the tools shown in Figure 4 were employed. The simulation used a two-dimensional axisymmetric model of an elastic-plastic material. The kinematic parameters governing the tool movement and the geometric conditions were carefully chosen to align with the current manufacturing process at CANPACK. To discretize the volume of the workpiece, we employed 6 -nodal triangular finite elements of the second order. With each computational Finite Element time step, the mesh was reconstructed, adapting to both the workpiece's geometry and the distribution of strain rate. The friction at the interface between the workpiece and the tool was specified using a combined friction law. This law incorporates not only the empirical friction factor but also factors such as the normal pressure at the contact and the flow stress value of the deformed material at the contact point. The friction factor's value, 0.15, was selected from the Qform database for conditions involving cold forging of aluminum alloys with mineral oil as a lubricant. In this context, a coupled problem was addressed, wherein the temperature distribution calculated at the ongoing time step influenced the mechanical problem through the modification of the flow stress of the deformed material.

The second scenario of calculations was aimed at determining the amount of internal pressure that would lead to the loss of stability of the can end (buckling strength tests). This calculation was necessary to evaluate the effect of temperature on the value of the critical internal pressure, leading to the loss of stability of the end. With the help of existing equipment, the solution of this problem experimentally is impossible. This problem was solved as a three-dimensional one, the type of elements used was a tetrahedron with a linear interpolation of the metal velocities. The internal pressure varied from zero to a value at which the product lost its mechanical stability.

In all computational categories for the simulation of large elastic-plastic deformations, Prandtl-Reuss equations were used [30]. Solution convergence was evaluated through rigorous parameters. In the context of the mechanical boundary problem, meticulous analysis was conducted on the flow velocity and average stress at discrete computational steps. For the heat flow problem, scrutiny was directed towards the temperature distribution. These convergence assessments held pivotal significance in guaranteeing the precision and dependability of the solution".

Point 2: In conclusion, an idea or guideline should be added for further research on the influence of temperature on the behavior of cans in production and storage.

Response 2: Thank you for your comments and suggestions. The following explanation has been added to the paper:

“Although our focus throughout this paper has been on the influence of temperature on the behavior of beverage ends in production and storage, the potential benefits of our findings are wider. Observed stamping force decrease may lead to some energy saving, and it can be worth effort as any calculation of final energy would be suitable to confirm this statement especially regarding to environment implications. These possibilities are yet to be researched.”

NOTE: All references in the response refer to figure numbers and references numbers for the modified manuscript.

With kind regards,

Yours sincerely,

Paweł Kokoszka

CANPACK S.A., Business Support Service, Starowiejska 28 Street, 32-800 Brzesko, Poland

E-mail: pawel.kokoszka@canpack.com

Round 2

Reviewer 2 Report

Recommended.

Minor.

Reviewer 3 Report

The addressed points have been answered. It is suggested to be accepted.